# Effect of Dispersed ZrO_2_ Particles on Microstructure Evolution and Superconducting Properties of Nb-Ti Alloy

**DOI:** 10.3390/ma17235946

**Published:** 2024-12-04

**Authors:** Rafał Idczak, Robert Konieczny, Wojciech Nowak, Wojciech Bartz, Michał Babij

**Affiliations:** 1Institute of Experimental Physics, University of Wrocław, pl. M. Borna 9, 50-204 Wrocław, Poland; robert.konieczny@uwr.edu.pl (R.K.); wojciech.nowak@uwr.edu.pl (W.N.); 2Institute of Low Temperature and Structure Research, Polish Academy of Sciences, ul. Okólna 2, 50-422 Wrocław, Poland; m.babij@intibs.pl; 3Institute of Geological Sciences, University of Wrocław, ul. Cybulskiego 30, 50-205 Wrocław, Poland; wojciech.bartz@uwr.edu.pl

**Keywords:** mechanical alloying, Nb-Ti, X-ray powder diffraction, superconductivity

## Abstract

The influence of dispersed ZrO_2_ particles on the microstructure evolution and the superconducting properties of a Nb-Ti alloy was investigated. The studied materials were prepared by different methods including mechanical alloying (MA) and arc-melting. The obtained samples were studied by X-ray diffraction (XRD) and vibrating-sample magnetometer (VSM). It was found that ZrO_2_ particles can be successively introduced into an Fe-Nb matrix by MA. However, among all prepared samples with a nominal composition of Nb-47wt%Ti-5 wt% ZrO_2_, only the powders, which were prepared by MA of Nb-47wt%Ti and ZrO_2_ powders, exhibit superconductivity with critical parameters comparable to those observed in pristine Nb-47wt%Ti alloy. In particular, the determined upper critical field at 0 K $\mu_{0}H_{c2}\left(0\right)$ is close to 15.6(1) T. This value is slightly higher than 15.3(3) T obtained for Nb-47wt%Ti and it can be ascribed to the presence of introduced ZrO_2_ particles in the Nb-Ti matrix.

## 1. Introduction

Oxide dispersion strengthening (ODS) is a method of improving the mechanical strength of certain materials by homogenously distributing thermally stable nano-sized oxides (i.e., Y_2_O_3_, ZrO_2_, SiO_2_, TiO_2_, Y_2_Ti_2_O_7_) in the matrix [1]. The fine dispersed particles can significantly reduce the motion of grain boundaries and dislocations, stabilize the microstructure at elevated temperature and increase the tensile and creep strength [2,3,4]. Since ODS alloys can withstand the extreme conditions, they are considered as high-performance structural materials for safe and sustainable use in gas turbine engines, hydrogen combustion power plants, advanced fusion and fission reactors. For that reason, in previous decades, ODS alloys have been intensively studied by experimental and theoretical methods of physics, chemistry, and material science [1,5,6].

ODS alloys are commonly fabricated by powder metallurgy methods, involving mechanical alloying (MA), which is a ball milling process where a powder mixture placed in the ball mill is subjected to high energy collision from the balls [7]. The technique of MA was originally developed by J. Benjamin to produce ODS superalloys in the middle of 1960s [7,8]. Today, MA is a well-known technique for synthesizing new materials including supersaturated solid solutions, metallic glasses, nanocrystalline materials, and high-entropy alloys [9].

In 2021, Mousavi et al. reported the investigation of dispersion of Y_2_O_3_ particles into the Nb-Ti superconductor using a processing strategy based on the ODS concept [10]. For the first time, the ODS material was prepared not to enhance its mechanical properties but in order to stabilize a superconducting state in the system. Following this idea, in this work, we present the study of influence of dispersed ZrO_2_ particles on microstructure evolution and superconducting properties of Nb-Ti alloy. The investigated materials were prepared by different methods including MA and arc-melting. The obtained samples were studied by X-ray diffraction (XRD) and vibrating-sample magnetometer (VSM) to characterize their phase composition and superconducting properties after each step of synthesis.

## 2. Materials and Methods

The samples with a nominal composition of Nb-47wt%Ti-5 wt% ZrO_2_ were prepared using MA in two series, powders A and B, and an additional sample C was prepared by arc-melting method. In the case of powders A, the mixture of Nb (99.99%, ∼325 mesh, Alfa Aesar, Heysham, UK), Ti (99.99%, ∼325 mesh, Alfa Aesar) and ZrO_2_ (99.99%, Sigma–Aldrich, Burlington, MA, USA) powders was placed in the grinding bowl with 25 grinding balls. The bowl and balls were made of hardened stainless steel. To avoid any contamination, the bowl was filled with argon. The milling was carried out at 430 rpm for effective times up to t=80 h. In the case of powders B, the Nb-Ti alloy was firstly prepared in an arc furnace using Nb sponges (99.8%, Thermal Scientific, Waltham, MA, USA) and Ti sponges (99.99%, Johnson Matthey, London, UK). In the next step, the obtained ingot was pulverised into fine powder and mixed with ZrO_2_ powder (99.99%, Sigma–Aldrich) before MA. Finally, the synthesis of sample C was conducted using an arc furnace using the same substrates as for powders B. The 1 g ingot of material was remelted four times to improve its homogeneity. The furnace chamber was filled with argon and pure titanium getter was melted before each melting of alloy ingot to purge the argon atmosphere of residual oxygen and nitrogen. After complete synthesis, the total weight loss of the obtained material did not exceed 0.2%.

The phase composition of the prepared materials was verified by powder X-ray diffraction (XRD) using a PANalytical X’pert Pro diffractometer with Cu Kα radiation ($\lambda_{1}=1.54056$ Å, $\lambda_{2}=1.54440$ Å). The measurements were carried out at room temperature. The experimental XRD patterns were analyzed with the FULLPROF software based on the Rietveld method [11,12]. Additionally, the mean grain size *L* and the level of internal stress Cϵ were estimated for each prepared sample using the Williamson–Hall method [13]:(1)$$\beta \cos\theta =C\epsilon \sin\theta +\frac{K\lambda }{L},$$
where β is the broadening of the diffraction peak at half its maximum intensity, λ corresponds to the wavelength of the X-ray beam used in the measurement, θ is the Bragg angle and K=0.9 denotes the dimensionless shape factor.

Measurements of dc and ac magnetic susceptibility as a function of temperature from 2 K to 15 K and applied magnetic field up to $\mu_{0}H$ = 9 T were performed using a quantum design Physical Properties Measurements System (PPMS) platform. The ac susceptibility measurements were conducted by applying an ac-field amplitude of $B_{\mathrm{ac}}$ = 1 mT with a frequency of *f* = 1 kHz. The presented values of the mass magnetic susceptibility $\chi_{m}$ as well as the real $\chi^{\prime }$ and imaginary $\chi^{\prime \prime }$ parts of ac-susceptibility have an uncertainty of about 5%.

The chemical composition and homogeneity of selected samples were verified by energy dispersive X-ray spectroscopy (EDXS) using a JEOL JSM-IT100 In-Touch-Scope^TM^ scanning electron microscope.

## 3. Results and Discussion

### 3.1. Chemical and Phase Composition

Figure 1 shows XRD patterns measured for powders A obtained after effective milling times up to t=80 h. The analysis of the experimental data performed using the Rietveld method reveals that the powders milled up to 40 h contain two phases: a body-centered cubic (bcc) structure (space group Im3¯m) and a hexagonal close-packed (hcp) structure (space group $P6_{3}/\mathrm{mmc}$). The lattice parameter *a* for bcc structure is close to 3.305(3) Å, which is a comparable value to a=3.306 Å determined for pure Nb [14] and slightly higher than a=3.286(1) Å obtained for Nb-47wt%Ti alloy prepared by arc-melting [15]. In the case of hcp structure, the lattice parameters a=2.949(1) Å and c=4.687(1) Å are in good agreement with corresponding values for high purity alpha titanium [16]. For the powders after 60 h and 80 h of MA, XRD patterns show an additional phase which is identified as a non-stoichiometric metallic carbide (probably NbTiC_x_), which crystallizes in rock–salt structure (space group $\mathrm{Fm}\bar{3}m$) with a=4.29(1) Å. In all XRD patterns measured for powders A, the diffraction peaks which could be ascribed to ZrO_2_ are not observed.

The phase composition of powders A and the *L* and Cϵ parameters determined for the bcc phase are presented in Figure 2. Taking into account XRD data, one can assume that during the MA up to 40 h, the Ti atoms as well as ZrO_2_ particles are successively introduced into the bcc Nb-rich phase. Longer milling times than 40 h result in carbon contamination of the sample and the formation of non-stoichiometric metallic carbides which crystallize in a rock–salt structure. Carbon contamination of the powder is most likely caused by mechanical alloying using stainless steel vial and balls. The values of *L* and Cϵ parameters determined for the bcc phase increase with the milling time. In the case of Cϵ parameter this result is expected, since MA leads to creation of many lattice defects which increase the mean level of internal strain in the powders. At first glance it seems the increase in mean grain size with the milling time is rather surprising since MA of many systems leads to a decrease in the particles’ size [9]. However, in the case of ductile–ductile systems such as Nb-Ti [17], Ag–Cu [18] or Ni–Ti [19], powder particles have tendency to weld together and form large grains.

Figure 3 shows XRD patterns measured for powders B obtained after effective milling times up to t=60 h. The analysis of the experimental data reveals that the powders milled up to 20 h contain only one bcc phase with lattice parameter close to 3.29(1) Å, which is comparable with that observed for bcc phase in powders A. For longer milling times, the presence of a non-stoichiometric metallic carbide NbTiC_x_ phase is detected.

As one can see in Figure 2, for t>20 h, the content of non-stoichiometric metallic carbide phase gradually increases with the milling time. Again, this effect is probably connected with the carbon contamination during the synthesis. The presence of metallic carbide is detected after *t* = 60 h for powders A and after 30 h for powders B. Therefore, one can conclude that the NbTiC_x_ phase is formed from bcc Nb-Ti alloy and carbon contamination. The changes in *L* and Cϵ parameters with the milling time (Figure 2) are similar to those observed for powders A. Both parameters increase with *t*. However, it can be noted that Cϵ estimated for powder B after 10 h of milling is more than twice higher than the corresponding one for powder A. This finding may be associated with arc-melting and subsequent powdering of the Nb-Ti sample before MA synthesis.

The chemical composition of two powders B after 20 and 60 h of MA were determined using SEM with EDX detector (see Figure 4 and Figure 5). It was found that powder B after 20 h of milling has the following chemical composition (in wt.%): 18.6(2.5)% C, 35.3(2.2)% Ti, 42.6(2.3)% Nb, 3.5(1.8)% Zr, while powder B after 60 h of MA is composed of 39.2(7.0)% C, 27.9(3.0)% Ti, 31.2(4.9)% Nb, 1.7(0.7)% Zr. As one can notice, the concentration of carbon drastically increases in studied powders with the milling time. Therefore, it is plausible to assume that the observed phase which crystallizes in rock–salt structure corresponds to non-stoichiometric metallic carbide (probably NbTiC_x_). The selected EDX spectra and the weight content of the studied samples are presented in the Appendix A.

### 3.2. Superconducting Properties

The XRD data collected for powders A and B reveal that the samples contain various phases. It should be noted here that pure Ti and ZrO_2_ particles do not exhibit superconductivity above 2 K. Therefore, the described below superconducting proprieties of the studied samples can be attributed to bcc Nb-rich phase and NbTiC_x_.

The selected temperature dependencies of mass magnetic susceptibility measured for powders A in the range of 2–15 K and in applied magnetic field up to $\mu_{0}H$ = 9 T are presented in Figure 6. As one can notice, $\chi_{m}\left(T\right)$ dependencies measured at low magnetic field of $\mu_{0}H$ = 10 mT reveal a strong diamagnetic response at low temperatures. This indicates that the studied powders undergo superconducting transition at critical temperature $T_{c}=8.9\left(2\right)$ K. The influence of the milling time on $T_{c}$ is negligibly small. Since the critical temperatures for high-purity Nb (9.25 K) [20] and Nb-47wt%Ti alloy (9.2 K) [15] are comparable to each other and only slightly higher than $T_{c}$ determined for all powders A, it is difficult to assess, using $T_{c}$, how many Ti atoms and ZrO_2_ particles were introduced into the Nb matrix during MA. Fortunately, pure Nb has a relatively low upper critical field $H_{c2}<1$ T [20], while $H_{c2}$ for the Nb-47wt%Ti alloy is close to 15 T [15]. As one can see in Figure 6, the $\chi_{m}\left(T\right)$ dependencies measured at $\mu_{0}H$ = 1 T clearly show that the superconducting diamagnetic signal increases with the increase in the milling time, indicating the formation of bcc alloy. At the same time, no strong diamagnetic signal was observed at $\mu_{0}H$ = 9 T. These results suggest that during MA, some Ti atoms and ZrO_2_ particles are successively introduced into the Nb matrix.

Real $\chi^{\prime }$ and imaginary $\chi^{\prime \prime }$ parts of the ac-susceptibility measured in applied magnetic field up to $\mu_{0}H$ = 9 T for powders A after 10 and 40 h of MA are presented in Figure 7 and Figure 8, respectively. The results obtained for powders A after 20 h and 30 h of milling are included in the Appendix A. These results provide a more detailed insight into the superconducting properties of the powders studied. In general, the diamagnetic behavior observed in $\chi^{\prime }$ confirms the presence of bulk superconductivity in the studied material. In the case of the imaginary part, the measured $\chi^{\prime \prime }$(T) curves reveal peaks that reflect energy dissipation and are characteristic of superconductors. As can be seen in Figure 7, the $\chi^{\prime \prime }\left(T\right)$ dependence measured in $\mu_{0}H$ = 1 mT shows two peaks that can be associated with the presence of two superconducting phases in this powder. The first narrow peak with maximum at 9.0(1) K is related to pure Nb since this peak is not observed in $\chi^{\prime \prime }\left(T\right)$ dependencies measured in $\mu_{0}H\geq$ 1 T. The second peak, which is broad and disappears only for magnetic fields greater than 2 T, can be connected with the mechanically alloyed bcc phase. For this phase, the onset of $T_{c}$ is close to 9.0(2) K in $\mu_{0}H$ = 1 mT and decreases to 4.0(2) K in $\mu_{0}H$ = 2 T. In the case of powder A milled for 40 h, one can observe that the $\chi^{\prime \prime }\left(T\right)$ peak assigned to pure Nb is relatively lower (see Figure 8) than for 10 h (see Figure 7). Again, this result suggests that during MA, some Ti atoms and ZrO_2_ particles are successively introduced into the Nb matrix. At the same time, the second peak in $\chi^{\prime \prime }\left(T\right)$ is much narrower than the one observed for powders milled for a shorter time. Moreover, the transition to the superconducting state is clearly visible in $\chi^{\prime }\left(T\right)$ and $\chi^{\prime \prime }\left(T\right)$ curves measured in $\mu_{0}H$ = 7 T with the onset of $T_{c}$ close to 3.5(2) K. This result suggests that longer grinding times have a positive effect on the superconducting properties of the prepared bcc alloy. Unfortunately, the superconducting critical parameters such as $T_{c}\approx$ 9 K and, especially, $\mu_{0}H_{c2}<$ 9 T estimated for the powder prepared by method A for 40 h are far below those determined for Nb-47wt%Ti alloy ($T_{c}$ = 9.2(1) K, $\mu_{0}H_{c2}$ = 15.3(3) T [15]). At the same time, the XRD data reveal that longer milling times than 40 h result in carbon contamination of the sample and the formation of non-stoichiometric carbides which crystallize in the rock–salt structure.

Figure 9 presents the selected temperature dependencies of mass magnetic susceptibility measured for powders B in the range of 2–15 K and in applied magnetic field up to $\mu_{0}H$ = 9 T. The $\chi_{m}\left(T\right)$ data obtained for powders B reveal that all these samples are superconductors with $T_{c}\approx$ 8.5 K in $\mu_{0}H$ = 10 mT. Moreover, the measurements performed in higher magnetic fields suggest that $\mu_{0}H_{c2}$ for these materials is higher than 9 T.

Real $\chi^{\prime }$ and imaginary $\chi^{\prime \prime }$ parts of the ac-susceptibility measured in applied magnetic field up to $\mu_{0}H$ = 9 T for powders B after 10, 40 and 60 h of MA are presented in Figure 10, Figure 11 and Figure 12, respectively. The results obtained for powders B after 20 h, 30 h and 50 h of milling are included in the Appendix A. The temperature dependencies of $\chi^{\prime }$ and $\chi^{\prime \prime }$ obtained for 10 h powder confirm that the sample is superconductor with $T_{c}\approx$ 9 K and $\mu_{0}H_{c2}>$ 9 T. The sharp and narrow peaks observed in $\chi^{\prime \prime }\left(T\right)$ dependencies for this powder indicate the good homogeneity of the superconducting state. With increasing milling time, the $\chi^{\prime \prime }\left(T\right)$ peaks become broader, which is presumably due to the structural disorder, introduction of ZrO_2_ particles into the Nb-Ti matrix and lattice strain introduced into the powder by MA. Surprisingly, in the case of powders B milled for 50 h and 60 h, the $\chi^{\prime }\left(T\right)$ and $\chi^{\prime \prime }\left(T\right)$ data reveal the presence of two superconducting phases. The basis for this statement is that two distinct peaks can be clearly seen in the $\chi^{\prime \prime }\left(T\right)$ curves. Moreover, the fact that both peaks gradually shift towards lower temperatures as the applied magnetic field increases additionally supports this statement. Taking into account the XRD results as well as the fact that NbC and TiC compounds are superconductors with $T_{c}$ close to 11 K [21] and 0.5 K [22], respectively, it is plausible to assume that the second superconducting phase in these powders is non-stoichiometric carbide crystallising in the rock–salt structure. In the case of powder B milled for 50 h, the $T_{c}$ of this carbide determined as the maximum of $\chi^{\prime \prime }\left(T\right)$ peak in $\mu_{0}H$ = 1 mT is close to 6.1 K, while for 60 h sample, $T_{c}$ decreases to 4.2 K. This finding suggests that longer milling times, which result in a higher carbon content in the carbide phase, have a negative effect on its superconducting properties.

The measurements of the magnetic properties of powders B indicate that for the superconducting bcc alloy, the upper critical field at 0 K ($\mu_{0}H_{c2}\left(0\right)$) is higher than 9 T. The Pauli limiting field $\mu_{0}H_{P}\approx$ 16.6 T was estimated from the equation [23]:(2)$$\mu_{0}H_{P}=1.84T_{c}.$$

Therefore, to obtain correct assessment of $\mu_{0}H_{c2}$ we applied the Werthamer–Helfand– Hohenberg (WHH) [24,25,26] model for BCS superconductors in the dirty limit, which includes the spin-paramagnetic effect (described by the Maki parameter $\alpha_{M}$) and the spin–orbit scattering constant $\lambda_{\mathrm{SO}}$ to fit the experimental data (Equation (Equation 3), where $\gamma \equiv \sqrt{{\left(\alpha_{M}\bar{h}\right)}^{2}-{\left(\frac{1}{2}\lambda_{\mathrm{SO}}\right)}^{2}}$, $\bar{h}=\frac{4}{\pi^{2}}\frac{H_{c2}}{-dH_{c2}/dT}$, $t\;=\;\frac{T}{T_{C}}$ and ψ is digamma function):(3)$$\begin{array}{c} \ln\frac{1}{t}=\left(\frac{1}{2}+\frac{i\lambda_{\mathrm{SO}}}{4\gamma }\right)\psi \left(\frac{1}{2}+\frac{\bar{h}+\frac{1}{2}\lambda_{\mathrm{SO}}+i\gamma }{2t}\right)+\\ \left(\frac{1}{2}-\frac{i\lambda_{\mathrm{SO}}}{4\gamma }\right)\psi \left(\frac{1}{2}+\frac{\bar{h}+\frac{1}{2}\lambda_{\mathrm{SO}}-i\gamma }{2t}\right)-\psi \left(\frac{1}{2}\right)\; \end{array}$$

The results are presented in Figure 13. As one can see, the values of $T_{c}$ and $H_{c2}$ determined for bcc phase in all powders B are comparable to each other, indicating the limited influence of milling time on superconducting properties of this phase. In other words, this finding suggests that the changes in the mean grain size as well as the level of internal strain observed for the bcc phase (see Figure 2) do not cause any noticeable changes in $T_{c}$ and $H_{c2}$ values. Due to that, it was decided to simulate only one set of WHH curves for all powders B. The best results were obtained for the Maki parameter $\alpha_{M}=1.40\left(3\right)$ and $\lambda_{\mathrm{SO}}=8\left(1\right)$, yielding $\mu_{0}H_{c2}\left(0\right)=15.6\left(1\right)$ T, $T_{c}$ = 9.0(1) K, and the orbital limiting field $\mu_{0}H_{c2}^{\mathrm{orb}}=16.5\left(1\right)$ T. Comparing these values with the results obtained for Nb-47wt%Ti alloy ($T_{c}$ = 9.2(1) K, $\mu_{0}H_{c2}\left(0\right)$ = 15.3(3) T [15]), one can notice that the influence of dispersed ZrO_2_ particles on the superconducting properties of the Nb-Ti alloy is rather small. However, it should be mentioned that $\mu_{0}H_{c2}\left(0\right)=15.6\left(1\right)$ T is slightly higher than 15.3(3) T obtained for Nb-47wt%Ti. Taking into account that the changes in *L* and Cϵ parameters do not cause any observable changes in $H_{c2}$ values, the increase in the upper critical field may be connected with the presence of introduced ZrO_2_ particles.

Figure 14 shows the temperature dependencies of mass magnetic susceptibility measured for sample C prepared by arc-melting in the range of 2–15 K and in applied magnetic field up to $\mu_{0}H$ = 9 T. As one can notice, in this sample superconductivity emerges at $T_{c}$ = 6.5(2) K, which is much lower value than that observed for bcc phase in powders A and B. This result can be explained by assuming that ZrO_2_ particles decompose to Zr and O atoms during the arc melting process. Despite the fact that the transition temperature of Nb-Ti alloy can be slightly increased by adding Zr atoms [27], the effect of interstitial oxygen on $T_{c}$ reduction may be dominant. According to works [28,29], the addition of interstitial oxygen atoms to niobium, in concentrations below the solubility limit, significantly reduces its transition temperature. In particular, $T_{c}$ drops by about 0.93 K for each 1 at.% O dissolved in Nb. Therefore, it is plausible to assume that the obtained sample C consists of the Nb-Ti alloy with the addition of substitutional metallic Zr solutes and interstitial oxygen atoms. At the same time, much higher $T_{c}$ observed for superconducting bcc phase in powders A and B indicates that during MA, the ZrO_2_ particles do not decompose to Zr and O, and they are introduced to the Nb-Ti matrix.

In summary, it was found that the sample A obtained after 40 h of MA of Nb, Ti and ZrO_2_ powders still contains pure Nb and Ti, while ZrO_2_ particles are successively introduced into the bcc Nb-Ti matrix. Unfortunately, the superconducting critical parameters such as $T_{c}\approx$ 9 K and, especially, $\mu_{0}H_{c2}<$ 9 T estimated for the powder prepared by method A for 40 h are far below those determined for Nb-47wt%Ti alloy ($T_{c}$ = 9.2(1) K, $\mu_{0}H_{c2}$ = 15.3(3) T [15]. At the same time, the XRD data reveal that longer milling times than 40 h result in carbon contamination of the sample and the formation of non-stoichiometric carbides which crystallize in rock-salt structure. In the case of powders B which were prepared by MA of Nb-47wt%Ti and ZrO_2_ powders, the bcc phase exhibits superconductivity below 9 K and the determined $\mu_{0}H_{c2}\left(0\right)=15.6\left(1\right)$ T is slightly higher than 15.3(3) T obtained for Nb-47wt%Ti. The increase in the upper critical field may be connected with the presence of introduced ZrO_2_ particles. In sample C, prepared by arc-melting procedure, the superconductivity emerges at $T_{c}$ = 6.5(2) K, which is much lower value than that observed for bcc phase in powders A and B. This result can be explained by assuming that ZrO_2_ particles decompose to Zr and O atoms during the arc melting process.

## 4. Conclusions

The influence of dispersed ZrO_2_ particles on microstructure evolution and superconducting properties of Nb-Ti alloy was investigated. The studied material with a nominal composition of Nb-47wt%Ti-5 wt% ZrO_2_ was prepared using MA in two series, powders A and B, and an additional sample C was prepared by arc-melting method. The obtained results reveal that the optimal method of synthesis of Nb-Ti alloy containing a dispersion of nanoscale ZrO_2_ particles is the powder metallurgy route by mechanical alloying Nb-47wt%Ti and ZrO_2_ powders using high energy ball milling. These powders contain a bcc phase which exhibits superconductivity with $T_{c}\approx$ 9 K and this value is comparable to $T_{c}$ = 9.2(1) K observed for pure Nb-47wt%Ti alloy. At the same time, the determined $\mu_{0}H_{c2}\left(0\right)=15.6\left(1\right)$ T is slightly higher than 15.3(3) T obtained for Nb-47wt%Ti. In the case of powders prepared using MA of pure Nb, Ti and ZrO_2_ particles as well as the sample obtained by arc-melting method, the superconducting critical parameters observed for bcc phase are much lower than those for powders B. Additionally, it was found that long milling times result in carbon contamination of the powders A and B, and the formation of non-stoichiometric metallic carbides which crystallize in a rock–salt structure. This contamination is originated from grinding bowl and grinding balls made of hardened stainless steel. It can be proposed, therefore, that the grinding tools for this type of synthesis should be made of a harder material, i.e., tungsten carbide.

## Figures and Tables

**Figure 1 materials-17-05946-f001:**
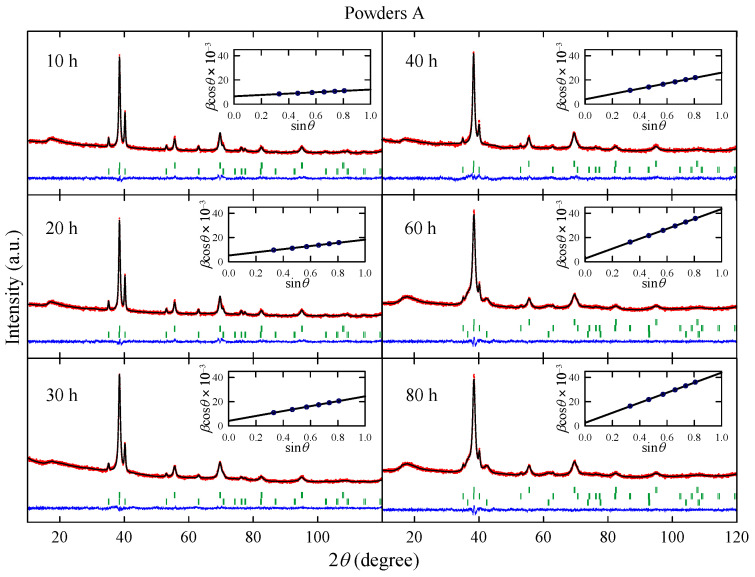
X-ray diffraction patterns measured for powders A obtained after effective milling times up to t=80 h. Red dots and black lines represent experimental data, and the result of the Rietveld refinement, respectively, and blue line shows the difference between the two. Green dashes indicate positions of the Bragg reflections. Insets show the Williamson–Hall plot; solid line is a linear fit to data points.

**Figure 2 materials-17-05946-f002:**
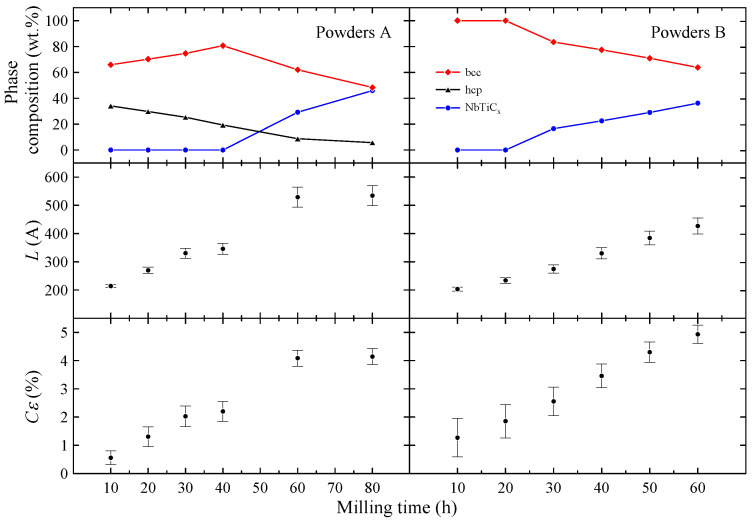
The phase composition of powders A and B. The estimated *L* and Cϵ parameters for the bcc phase.

**Figure 3 materials-17-05946-f003:**
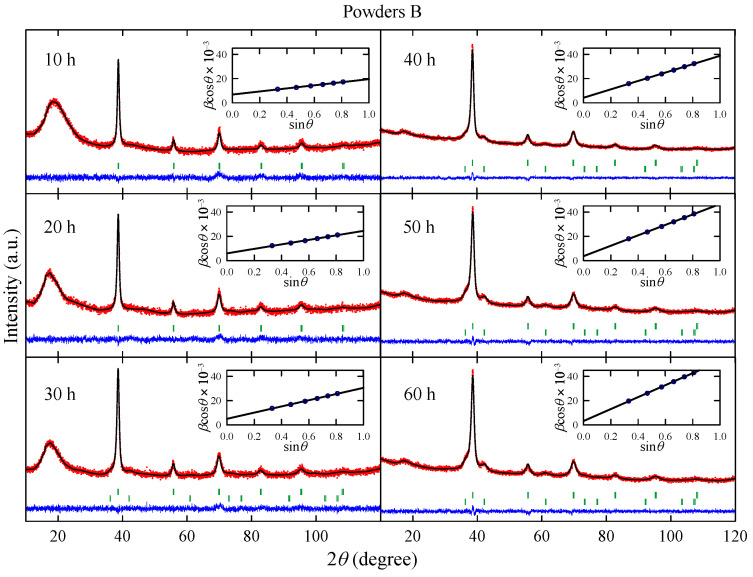
X-ray diffraction patterns measured for powders B obtained after effective milling times up to t=60 h. Red dots and black lines represent experimental data, and the result of the Rietveld refinement, respectively, and blue line shows the difference between the two. Green dashes indicate positions of the Bragg reflections. Insets show the Williamson–Hall plot; a solid line is a linear fit to data points.

**Figure 4 materials-17-05946-f004:**
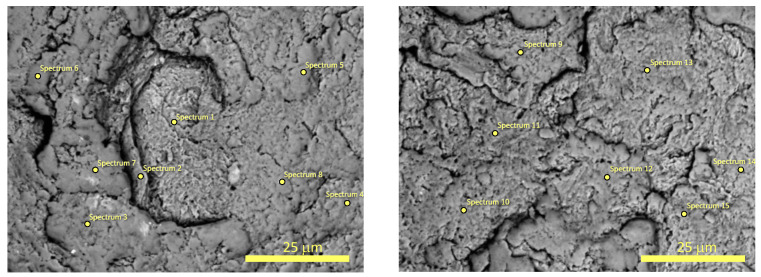
The SEM micrographs with marked points for EDX microanalysis of the surfaces of the powder B after 20 h of MA.

**Figure 5 materials-17-05946-f005:**
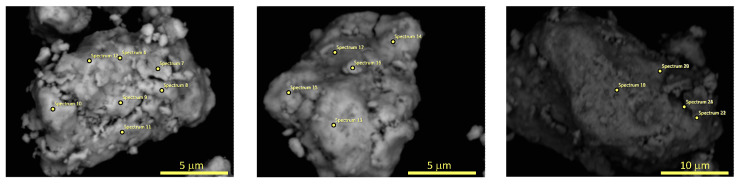
The SEM micrographs with marked points for EDX microanalysis of the surfaces of the powder B after 60 h of MA.

**Figure 6 materials-17-05946-f006:**
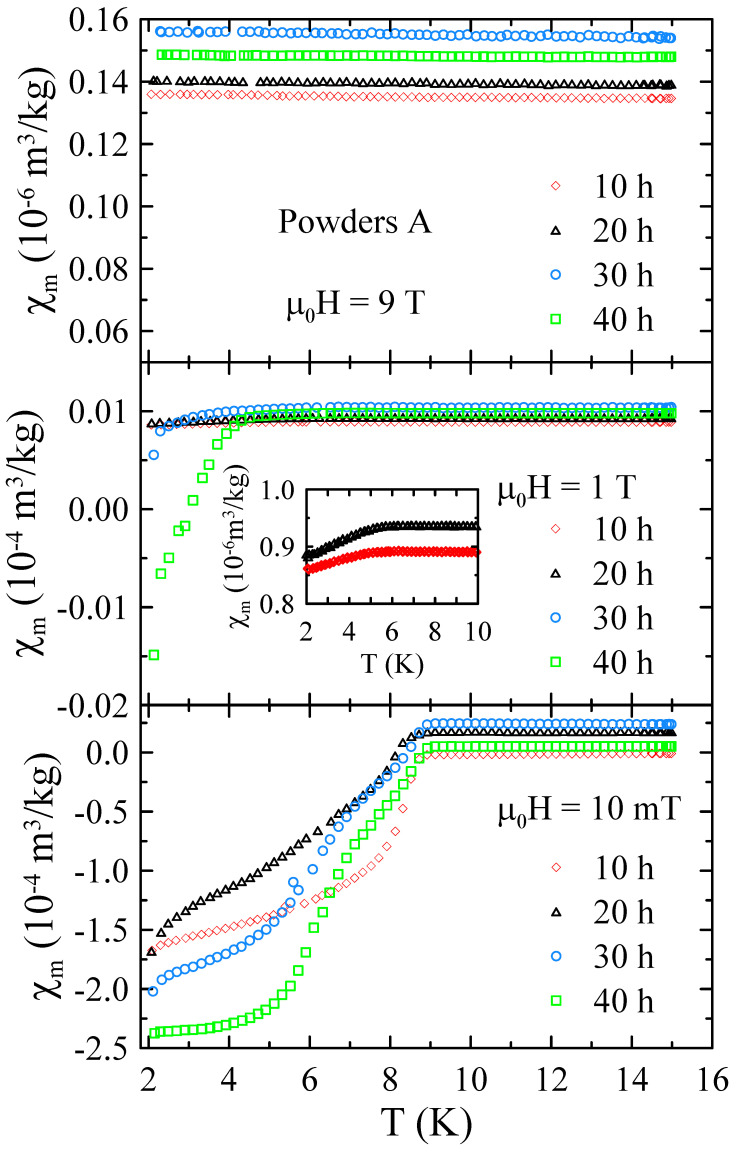
The selected temperature dependencies of mass magnetic susceptibility measured for powders A in the range of 2–15 K and in applied magnetic field up to $\mu_{0}H$ = 9 T.

**Figure 7 materials-17-05946-f007:**
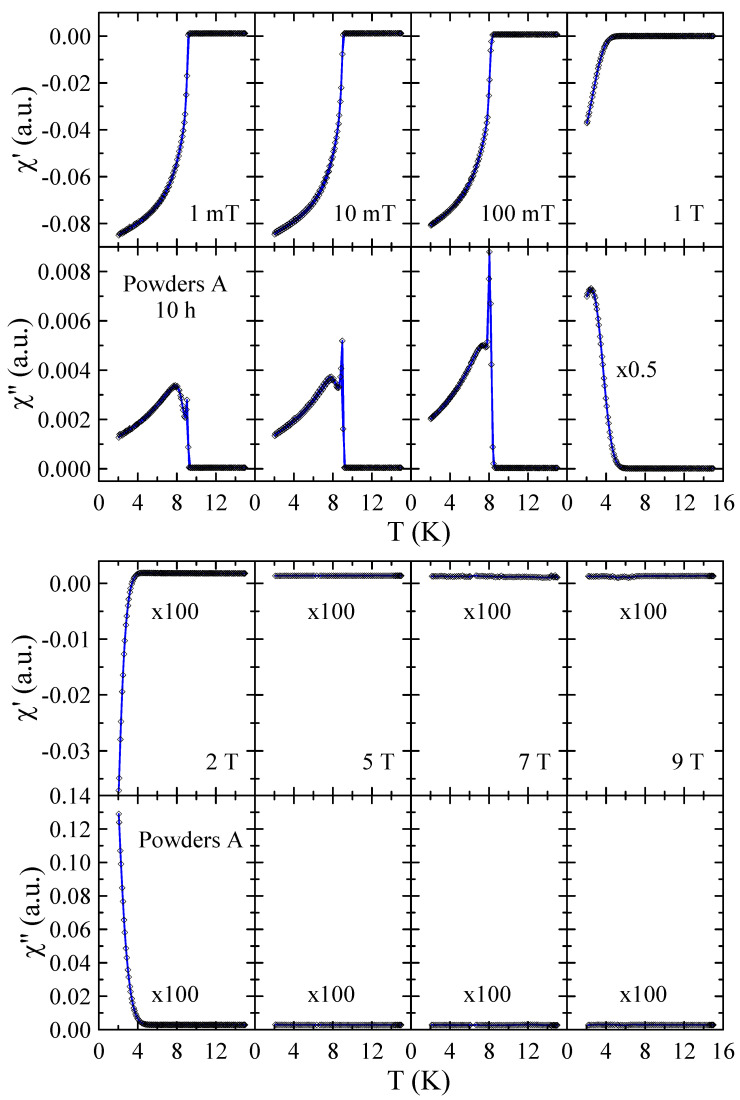
Real $\chi^{\prime }$ and imaginary $\chi^{\prime \prime }$ parts of the ac-susceptibility measured in applied magnetic field up to $\mu_{0}H$ = 9 T for powder A after 10 h of MA. Solid curves serve as guides for the eye.

**Figure 8 materials-17-05946-f008:**
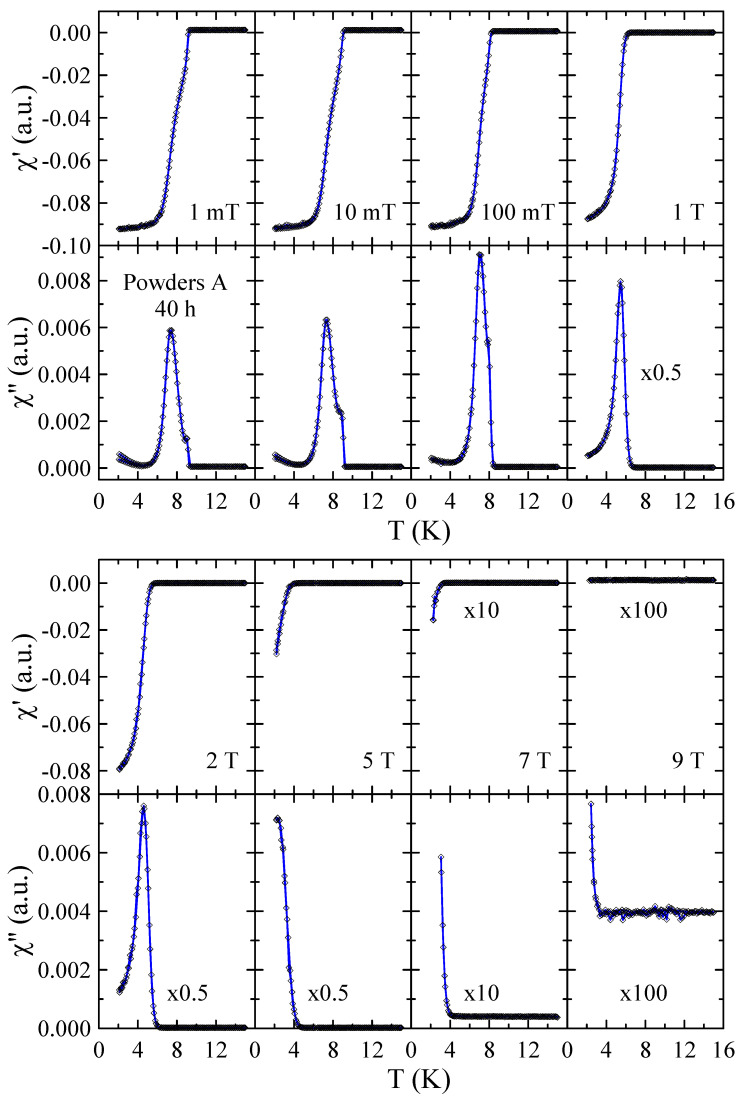
Real $\chi^{\prime }$ and imaginary $\chi^{\prime \prime }$ parts of the ac-susceptibility measured in applied magnetic field up to $\mu_{0}H$ = 9 T for powder A after 40 h of MA. Solid curves serve as guides for the eye.

**Figure 9 materials-17-05946-f009:**
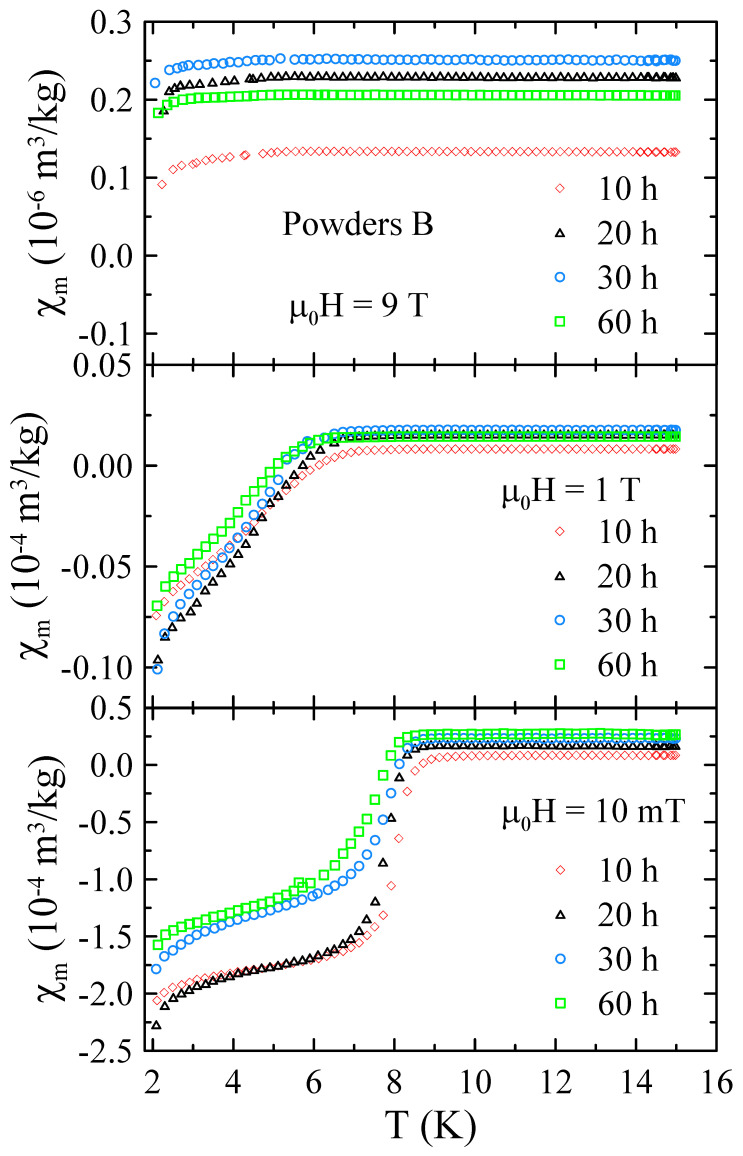
The selected temperature dependencies of mass magnetic susceptibility measured for powders B in the range of 2–15 K and in applied magnetic field up to $\mu_{0}H$ = 9 T.

**Figure 10 materials-17-05946-f010:**
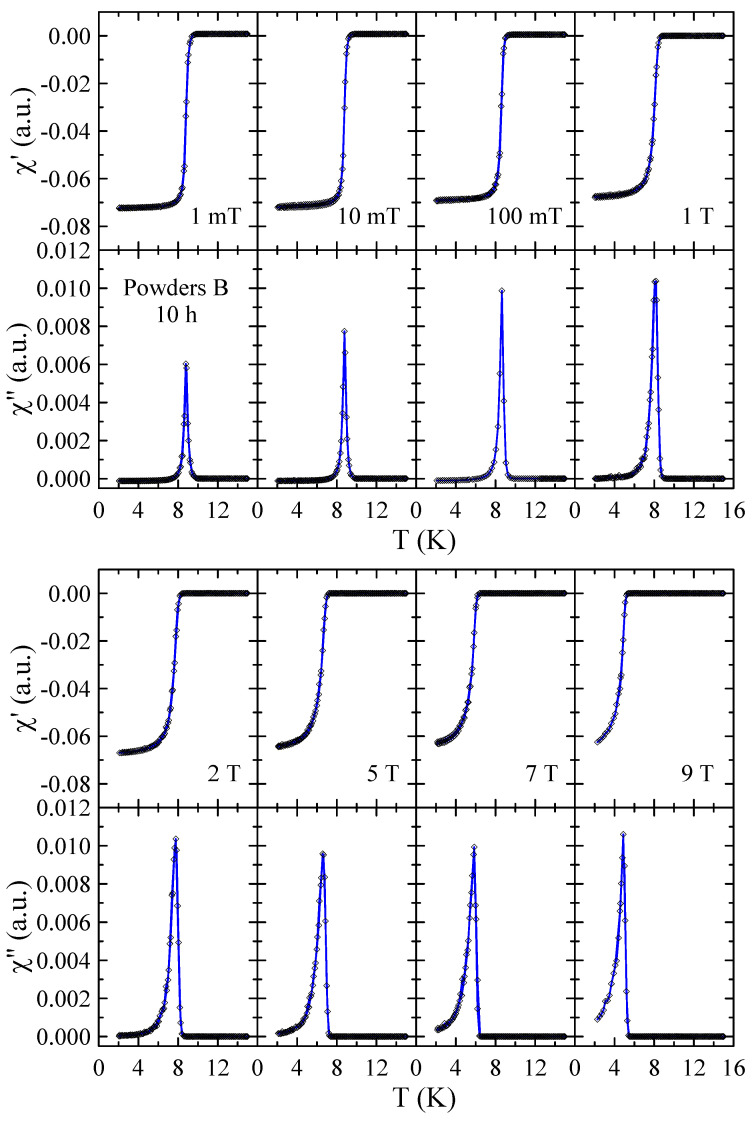
Real $\chi^{\prime }$ and imaginary $\chi^{\prime \prime }$ parts of the ac-susceptibility measured in applied magnetic field up to $\mu_{0}H$ = 9 T for powder B after 10 h of MA. Solid curves serve as guides for the eye.

**Figure 11 materials-17-05946-f011:**
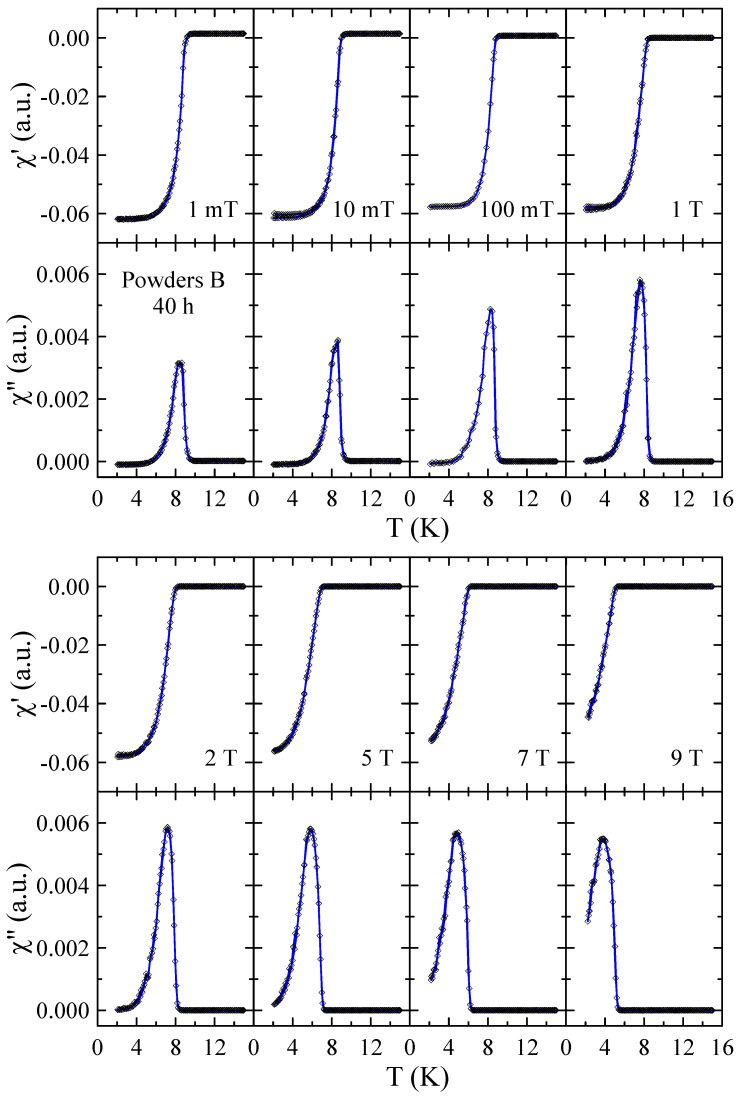
Real $\chi^{\prime }$ and imaginary $\chi^{\prime \prime }$ parts of the ac-susceptibility measured in applied magnetic field up to $\mu_{0}H$ = 9 T for powder B after 40 h of MA. Solid curves serve as guides for the eye.

**Figure 12 materials-17-05946-f012:**
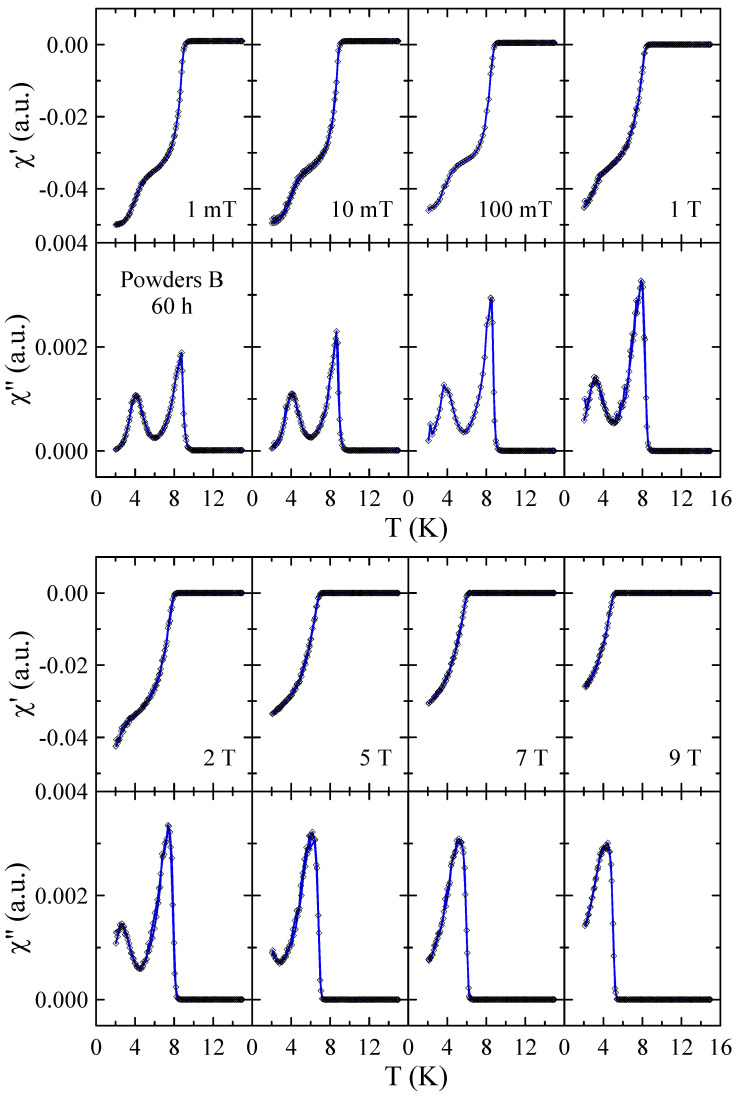
Real $\chi^{\prime }$ and imaginary $\chi^{\prime \prime }$ parts of the ac-susceptibility measured in applied magnetic field up to $\mu_{0}H$ = 9 T for powder B after 60 h of MA. Solid curves serve as guides for the eye.

**Figure 13 materials-17-05946-f013:**
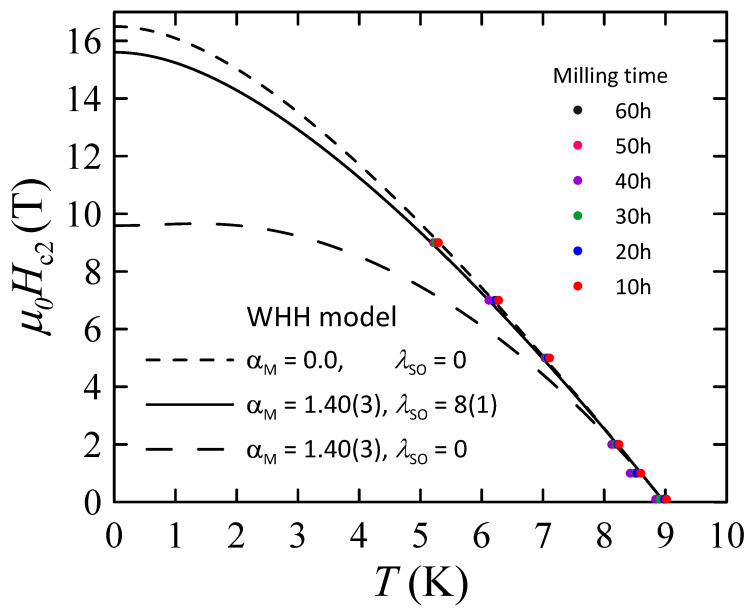
Upper critical field of superconducting bcc alloy in powders B as a function of temperature and derived from ac-susceptibility measurements. Solid and dashed lines are simulated WHH curves.

**Figure 14 materials-17-05946-f014:**
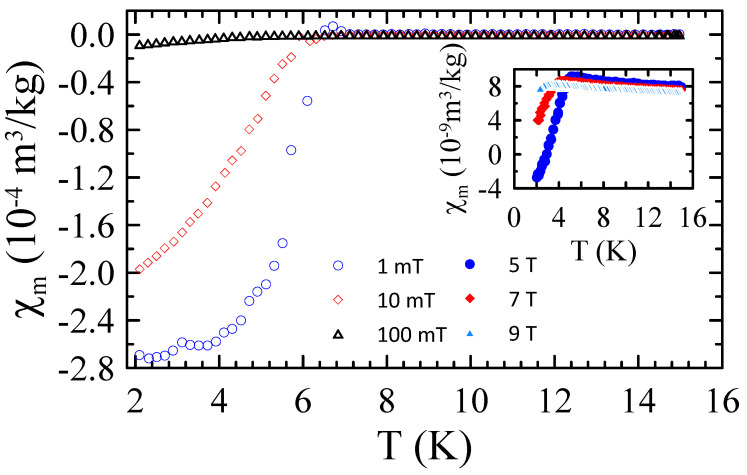
The temperature dependencies of mass magnetic susceptibility measured for sample C prepared by arc-melting in the range of 2–15 K and in applied magnetic field up to $\mu_{0}H$ = 9 T.

## Data Availability

Dataset available on request from the authors.

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
