# Peer review of "Effect of Dispersed ZrO_2_ Particles on Microstructure Evolution and Superconducting Properties of Nb-Ti Alloy"

_materials, 2024, doi:10.3390/ma17235946_

Round 1
Reviewer 1 Report
Comments and Suggestions for Authors
This paper investigates the influence of dispersed ZrO2 particles on the microstructural evolution and superconducting properties of Nb-Ti alloys. Different methods were used by the authors to prepare their materials, including mechanical alloying (MA) and arc melting. Samples were examined using X-ray diffraction (XRD) and vibrating sample magnetometers (VSM. According to their results, ZrO2 particles can be continuously introduced into Fe-Nb matrix using MA, however, of all samples produced with a nominal composition of Nb-47 wt% Ti-5 wt% ZrO2, only a powder prepared using MA from 6 Nb-47 wt% Ti and ZrO2 powders exhibits superconductivity with critical parameters comparable to those of the original alloy. At 0 K, they estimated an upper critical magnetic field of 15.6 T due to the presence of ZrO2 particles in the Nb-Ti matrix.
Despite the good experimental results, this manuscript has some shortcomings that can be summarized as follows:
-There seems to be an error since the references are not numbered and instead of numbers, there are interrogation symbols. Furthermore, no reference is mentioned at the end of the manuscript.
-In the introduction: μ0Hc2(0) is close to 15.6(1). The units of μ0Hc2(0) must be specified
-To determine the internal stress parameter Ce (eq. 1), the authors used the Williamson–Hall method. They used the parameter K=0.9. Can they justify this choice?
-The extrapolation estimate of Hc2 to 0K, (Fig. 13), is highly questionable since the authors did not measure below 4.2K. In addition, the curve of Hc2(T) is not linear. It is therefore very unfair to do such an extrapolation at zero K. They want Hc2 to be as high as possible, at any price. It’s unfair, we have to be reasonable.
In addition, equation (3) used for the adjustment contains many uncontrolled parameters.
-It is also not known where the experimental values of Hc2 used in the figure 13 for fit came from. They should be deduced from M(H) at different temperatures.
-Figures 5-14 must be reinserted in the heart of the text at their proper places
-References are not listed in the manuscript.
Taking into account the above remarks, I think that the manuscript cannot be accepted for publication in the current form.
Author Response
We thank the Referee for numerous comments. In all cases they were taken into account.
-There seems to be an error since the references are not numbered and instead of numbers, there are interrogation symbols. Furthermore, no reference is mentioned at the end of the manuscript.
Discussion: Corrected
-In the introduction: μ0Hc2(0) is close to 15.6(1). The units of μ0Hc2(0) must be specified
Discussion: Corrected
-To determine the internal stress parameter Ce (eq. 1), the authors used the Williamson–Hall method. They used the parameter K=0.9. Can they justify this choice?
Discussion: The shape factor depends on the space group of the XRD pattern analyzed and on the Miller indices of the individual reflections. Typically K value is in the range of 0.8-1.0. In our study, the main aim was to compare samples with each other, so a global shape factor value for the main phase was assumed to simplify the analysis.
-The extrapolation estimate of Hc2 to 0K, (Fig. 13), is highly questionable since the authors did not measure below 4.2K. In addition, the curve of Hc2(T) is not linear. It is therefore very unfair to do such an extrapolation at zero K. They want Hc2 to be as high as possible, at any price. It’s unfair, we have to be reasonable.
In addition, equation (3) used for the adjustment contains many uncontrolled parameters.
Discussion: The WHH model is well known physical model which describes μ0Hc2(T) dependence in superconductors. For more than 50 years this model is used to fit the low temperature part of the Hc2 data giving Hc2(0).
Equation 3 represents a convenient form of the WHH model. If we introduce the parameters γ, h ̅ and t, we obtain the explicit form of this equation, the solution of which is the function μ0Hc2(T), which depends only on Tc, αM and λSO. It is worth mentioning that the parameters Tc and αM result from a linear regression performed on the points measured at the highest temperatures. Thus, the only fitting parameter remains λSO.
-It is also not known where the experimental values of Hc2 used in the figure 13 for fit came from. They should be deduced from M(H) at different temperatures.
Discussion: The temperature of the superconducting transition in the applied magnetic fields was determined from χ’’(T) data taken in various magnetic fields. In our opinion this method is comparable and even more reliable than the determination of Hc2 values from M(H) dependences at different temperatures. In particular, the M(H) dependences taken at high magnetic field may contain paramagnetic component which makes it very difficult to determine the value of Hc2.
-Figures 5-14 must be reinserted in the heart of the text at their proper places
Discussion: Corrected
-References are not listed in the manuscript.
Discussion: Corrected
Reviewer 2 Report
Comments and Suggestions for Authors
Please find attached my review of the article.

Author Response
We thank the Referee for numerous comments. In all cases they were taken into account.
- There is no “References” section and the references themselves have no numbering, which makes the article difficult to review. This must to be revised.
Discussion: Corrected
- I recommend that all figures be placed close to where they are mentioned for the first time in the text. Currently the figures are spread over the article in such a way that makes it difficult to read it properly.
Discussion: Corrected
- Figures 3, 4 and 5 are in section 3.2, however that section does not discuss those figures, which leads to confusion. As mentioned in point 2, move them close to where they appear in the text for the first time.
Discussion: Corrected
- Regarding Figures 4 and 5, please add either a table with the elemental % or a diagram of the EDX measurements. Also please mark more clearly the points for EDX microanalysis.
Discussion: The points for EDX microanalysis are marked more clearly in Figs. 4 and 5. The table with the elemental % for selected EDX spectra is presented in supplementary materials.
- Please include the error % of the EDX apparatus that you used. On rows 127-128 it is mentioned "Finally, it should be mentioned that the Zr content in both powders is below the detection limit of measuring instrument." - it is important to know.
Discussion: This sentence has been corrected. We add the information about Zr content in both studied samples determined from EDX measurements.
- I recommend that the authors include a summarization in the end of discussion for the results and how they are influenced by each method used to synthesize the powders.
Discussion: The summarization in the end of discussion has been added.
- Do the authors have results for bulk NbTi samples and if they do - please add a short comparison between them and the results from this article.
Discussion: The comparison between structural and superconducting properties of bulk Nb-47wt.%Ti alloy and the studied in this work materials are presented in the manuscript.
Round 2
Reviewer 1 Report
Comments and Suggestions for Authors The authors have taken into account the comments and suggestions made. Although discussions about the model are ongoing and remain open, I now give a favorable opinion for the publication of the paper.